# Detection of Eucalyptus Leaf Disease with UAV Multispectral Imagery

Kuo Liao [1], Fan Yang [2,3], Haofei Dang [1], Yunzhong Wu [4], Kunfa Luo [4] and Guiying Li [2,3,*]

[1]  Wuyi Mountain National Meteorological Observation Station, Wuyishan 354200, China
[2]  Fujian Provincial Key Laboratory for Subtropical Resources and Environment, Fujian Normal University, Fuzhou 350007, China
[3]  Institute of Geography, Fujian Normal University, Fuzhou 350007, China
[4]  Yuanling State Forestry Farm, Yunxiao County, Zhangzhou 363300, China
[*]  Correspondence: liguiying@fjnu.edu.cn

**Abstract:** Forest disease is one of the most important factors affecting tree growth and product quality, reducing economic values of forest ecosystem goods and services. In order to prevent and control forest diseases, accurate detection in a timely manner is essential. Unmanned aerial vehicles (UAVs) are becoming an important tool for acquiring multispectral imagery, but have not been extensively used for detection of forest diseases. This research project selected a eucalyptus forest as a case study to explore the performance of leaf disease detection using high spatial resolution multispectral imagery that had been acquired by UAVs. The key variables sensitive to eucalyptus leaf diseases, including spectral bands and vegetation indices, were identified by using a mutual information–based feature selection method, then distinguishing disease levels using random forest and spectral angle mapper approaches. The results show that green, red edge, and near-infrared wavelengths, nitrogen reflectance index, and greenness index are sensitive to forest diseases. The random forest classifier, based on a combination of sensitive spectral bands (green, red edge, and near-infrared wavelengths) and a nitrogen reflectance index, provided the best differentiation results for healthy and three disease severity levels (mild, moderate, and severe) with overall accuracy of 90.1% and kappa coefficient of 0.87. This research provides a new way to detect eucalyptus leaf diseases, and the proposed method may be suitable for other forest types.

**Keywords:** eucalyptus diseases; mutual information–based feature selection; random forest; spectral angle mapper; vegetation index

## 1. Introduction

Forest disease is one of the important factors stunting tree growth and reducing wood product quality and economic values of forest ecosystem goods and services [1,2]. Conventional detection of forest diseases is based on field surveys. A survey can accurately quantify disease infection stages and damage levels, but is costly and time consuming [3]. Moreover, it is hard for investigators to reach remote areas, especially where terrains and forest compositions are complex, leading to low efficiency and challenges for early warning systems and for the forecasts of forest diseases [4,5]. Remote sensing provides a new way to monitor forest health conditions at a reduced cost over large areas.

Remotely sensed data with various spatial, spectral, and temporal resolutions can be collected by diverse platforms and sensors, facilitating rapid and accurate monitoring of forest ecosystem functions and detection of disturbances caused by human activities and natural disasters (e.g., fire, flooding, drought, and disease/pests) at different scales, even in remote and complex forested areas [3,6,7]. In addition, the capability of remote sensing using wide ranges of electromagnetic spectra, especially infrared wavelength, enables it to detect subtle changes in tree or forest properties that are invisible to human eyes. The theory of remote sensing–based forest disease detection is that when trees are infected by

pathogens, the consequent changes in biophysical, physiological, biochemical, structural, and functional characteristics are reflected in specific wavelengths of the electromagnetic spectra, which are recorded by sensors [3]. For example, a change in the cell structure of leaves decreases reflectance values in the near-infrared (NIR) wavelength [8], while the loss of water content in the foliar increases reflectance values in the shortwave infrared (SWIR) wavelength [9].

Previous research has examined different kinds of remotely sensed data including optical [10,11], radar [12], Lidar [13], and thermal [14,15] for detecting forest diseases, and major methods have been reviewed [1,3,7,16]. At present, optical imagery (multispectral and hyperspectral) is still a major data source for detecting and monitoring forest diseases. Previous studies have identified a large number of spectral, textural, and structural features derived from optical imagery for differentiating healthy and damaged trees [7]. For example, red and red edge spectral bands of Sentinel-2 multispectral imagery were found to be the most important in discriminating coffee leaf rust levels [17], while green and red edge bands were found to be sensitive to eucalyptus damage caused by the Cossid Moth [18], and red edge and NIR bands of WorldView-2 multispectral imagery were the most important for detection of eucalyptus damage caused by the bronze bug [19].

Vegetation indices can extract or enhance some new features from combinations of spectral bands that individual bands do not have and outperform the use of spectral bands alone in assessing plant conditions [20]. For instance, the ratio of middle infrared band to NIR band of Landsat Thematic Mapper data was highly responsive to the damages caused by the Pear Thrips in spruce and caterpillars on Masson pine [21,22]; plant senescence reflectance index, normalized difference water index, moisture stress index, and enhanced vegetation index derived from EO-1 Hyperion were identified to be the most effective in assessing spruce budworm defoliation [23]. It is worth noting that red edge and red edge-based vegetation indices have advantages in assessing and monitoring vegetation stress because of their high sensitivity to chlorophyll content in vegetation. For example, red edge-based vegetation indices from RapidEye imagery were found to outperform other vegetation indices (e.g., NDVI) in assessing forest disturbance in Norway spruce–dominated forests in mountainous areas [24].

Experimental studies showed that spatial resolution of remotely sensed data plays an important role in classifying healthy and infected vegetation [25,26]. For instance, the accuracy of detecting healthy and infected Norway Spruce based on WorldView-2 data with a spatial resolution of 1.8 m could reach 75% [27], while it was only 64% based on hyperspectral HyMAP data with a spatial resolution of 4 m [28]. However, high-resolution spaceborne images are acquired at a fixed time intervals and heavily rely on weather conditions. The unmanned aerial vehicle (UAV) provides an alternative means for forest disease detection and monitoring [29,30]. UAVs have many advantages over satellite sensors, including high flexibility, fast response, and high spatial resolution. The advanced high–spatial resolution data obtained by UAVs can capture subtle changes in trees caused by diseases, and are increasingly applied to forest disease detection and monitoring [2,29,31–33]. For example, Näsi et al. [34] identified infected European spruce trees and classified them into three health conditions (healthy, infected by spruce bark beetle, and dead) with overall detection accuracy of 81% using a support vector machine (SVM) algorithm based on the UAV hyperspectral data; Wu et al. [35] analyzed the performance of regular machine learning classifiers and deep-learning object detection models for early diagnosis of pine wilt disease using large numbers of UAV RGB (red, green, blue) images, and found that a random forest algorithm yielded the best classification results. Similar results were also obtained by Yu et al. [36] based on UAV multispectral data.

Eucalyptus, due to its high adaptability and productivity, fast growth, short investment cycle, and high economic values of wood products, has become the most widely cultivated species in southern China [37]. However, monocultured plantations are prone to diseases, which are stunting growth and affecting timber quality. The common eucalyptus diseases in southern China include bacterial wilt caused by Ralstonia solanacearum, stem

diseases, and leaf diseases caused by various fungi [37]. Evidence shows that disease occurrence and severity in eucalyptus plantations have increased in recent years [38,39]. Chinese researchers have developed various treatments and management techniques to prevent and control eucalyptus diseases, including pruning or removing infected parts or whole trees, spraying with pesticides, introducing biological enemies, improving soil conditions, and planting disease-resistant clones [37,40]. Accurate detection of infected trees and assessment of infection severity are prerequisites for implementing those measures promptly. Remote sensing has long been used to detect eucalyptus crown damage caused by insects and diseases [41–44]. The Eucalypt Canopy Condition Index was specifically developed for evaluation of eucalyptus health conditions using high-resolution spatial and spectral remote-sensing images [42,45,46]. There are also some efforts using medium spatial resolution remote-sensing data, such as Landsat, Hyperion, and Sentinel-2, in monitoring eucalyptus health conditions in a large area [18,47,48].

Considering the difficulty in detecting infected trees using relatively coarse spatial resolution, application of UAV-collected data in disease detection and damage assessment of eucalyptus plantations has increased, and most of these studies have used RGB data [5,49,50], while limited cases have used multispectral [51] and thermal data [52]. Although UAV technology is regarded as a valuable tool for detecting forest diseases, rarely has research been explored for eucalyptus tree diseases in China. This research selected Yuanling Forest Farm, Yunxiao County as a case study to explore the capability of UAV multispectral images in detecting infected eucalyptus trees, and to identify the optimal features and algorithms to distinguish different infection severity levels. The results are expected to provide scientific support for eucalyptus disease early warning, and prompt implementations of measures to control the pathogen's spread to surrounding susceptible trees.

## 2. Materials and Methods

### 2.1. Study Area

The study area is located in Yuanling Forest Farm, Yunxiao County, Fujian Province (Figure 1). Covering 1054.3 km$^2$, Yunxiao County sits on the coast in southern Fujian Province. It is surrounded by hilly terrain to the north, west, and south, but the central and eastern coast are flat with alluvial plains and terraces. The climate is southern subtropical marine monsoon, with annual average temperature of 21.2 °C and annual precipitation of 1730.6 mm. The climate and soil conditions are favorable for eucalyptus growth. Because of the short rotation cycle and high economic return, eucalyptus plantations have expanded vastly in Yunxiao County during the past two decades. However, there were reports of frequent occurrences of diseases in these monoculture plantations, mainly leaf diseases called red leaf dieback, severely affecting eucalyptus growth, which has raised considerably the awareness of researchers and farmers.

Considering disease occurrences during recent years and topographic features, the experimental site covering about 1 km$^2$ in Yuanling Forest Farm (Figure 1) was selected. Eucalyptus plantations account for about one-third of the total experimental area. Average density of plantations is 1650 trees per hectare. Other major tree species include lychee, schima, and pine.

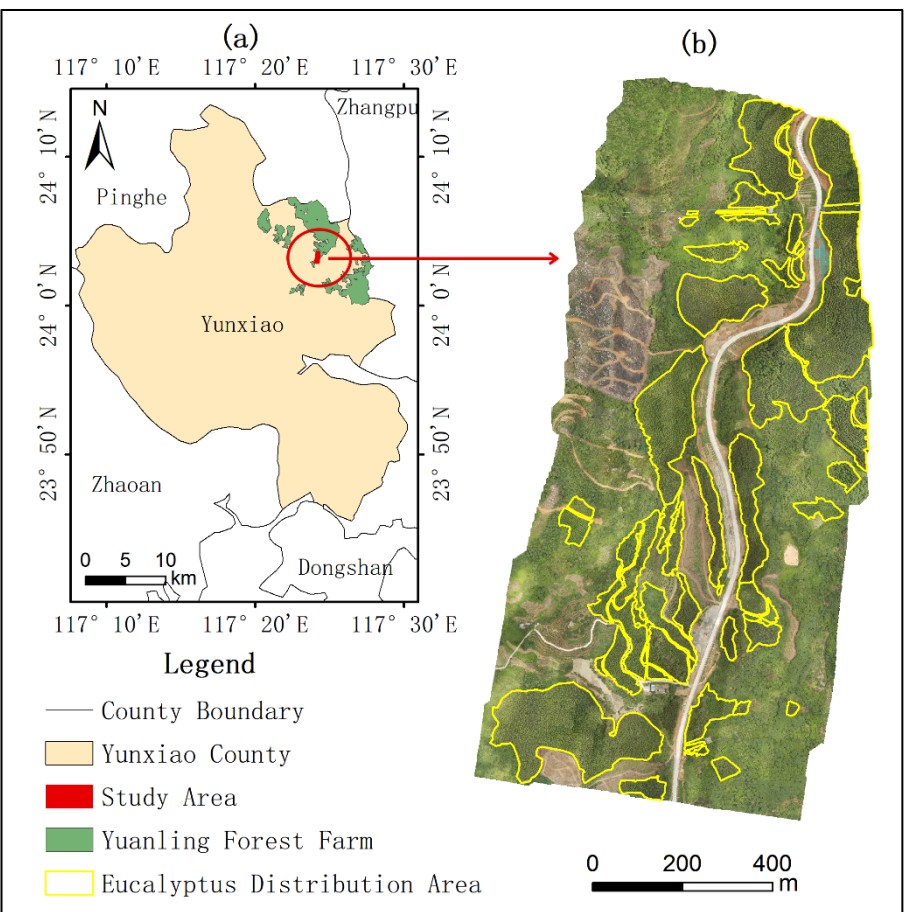

**Figure 1.** Location of study area. (**a**) Yuanling Forest Farm, Yunxiao County, Fujian Province, China; (**b**) UAV multispectral imagery, superposed by eucalyptus plantations.

*2.2. Study Framework*

The objective of this study is to explore the capability of UAV multispectral data for detecting infected eucalyptus trees. Figure 2 illustrates the framework of the research. It consists of the following major steps: (1) collection of data, including UAV multispectral images and samples of eucalyptus tree health conditions; (2) identification of optimal spectral bands and vegetation indices (VIs) using the mutual information–based feature selection method (MI); (3) detection of infected trees using Random Forest (RF) and Spectral Angle Mapper (SAM) based on the selected optimal spectral bands, and their combination with VIs; (4) determination of optimal features and algorithm through accuracy assessment and comparison of results.

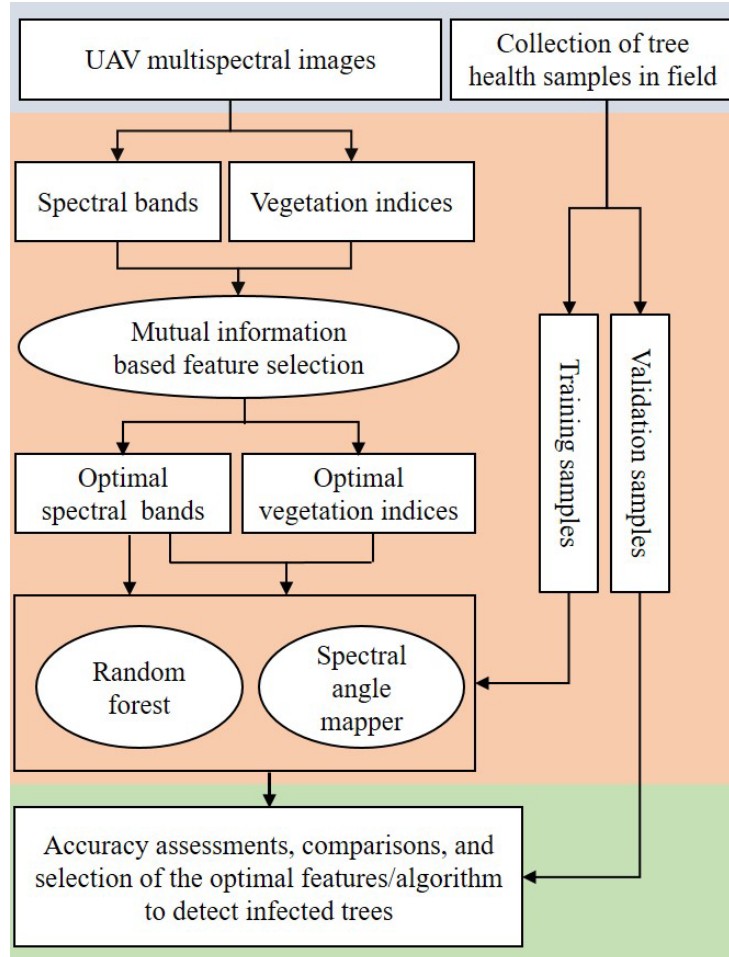

**Figure 2.** The framework for detecting diseased eucalyptus trees based on UAV multispectral data.

*2.3. Data Collection*

2.3.1. UAV Multispectral Data

UAV multispectral images were acquired on 21 July 2021, under clear weather condition using a DJI Phantom 4 Multispectral. The DJI Phantom 4 carries one RGB camera and five multispectral cameras covering blue ($450 \pm 16$ nm), green ($560 \pm 16$ nm), red ($650 \pm 16$ nm), red edge ($730 \pm 16$ nm), and NIR bands ($840 \pm 26$ nm). The flight parameters were set at a height of 180 m, speed of 9 m/s, exposure interval of 2 s, and side overlap of 85%; spatial ground resolution was about 0.1 m. The images were mosaicked, and a digital orthophoto map (DOM) with spatial resolution of 0.1 m was generated using DJI Terra software.

2.3.2. Eucalyptus Tree Samples of Health Status

A field survey for eucalyptus tree health samples was conducted in July 2021, the same day as the UAV data acquisition. Infected eucalyptus trees have distinctive symptoms, compared to healthy ones, that exhibit in leaf color—bright yellow, carmine, brown, and grey, depending upon the severity of damage. According to the symptoms in leaf colors and the amount of unhealthy leaves on trees observed during the field survey, four categories of health levels were defined: healthy, mildly infected, moderately infected, and severely infected. The infected trees with less than 30% of leaves turning to yellow or dark belong to the mild category, and the infected trees with greater than 60% of leaves turning to yellow or dark, as well as the defoliated or dead trees, fall into the severe category, while the ones between mild and severe are moderate. The infected trees demonstrate spectral responses distinguishable from those of the healthy trees on UAV multispectral images.

Figure 3 illustrates the typical samples of healthy trees and three infection levels on UAV multispectral images. Over the UAV experimental site, a total of 303 tree samples were collected in the field by visual observation, including 100 healthy trees, 50 mildly infected trees, 73 moderately infected trees, and 80 severely infected trees. Most infected trees were planted in 2013–2014, and their average crown diameter was about 3.5 m. The geophysical positions of those trees were recorded by GPS and superposed on UAV multispectral images; sample polygons were then drawn manually around tree crowns. The stratification-sampling method was applied to divide the sample trees into two portions: 70% as the training samples for image classification, and 30% for accuracy assessment.

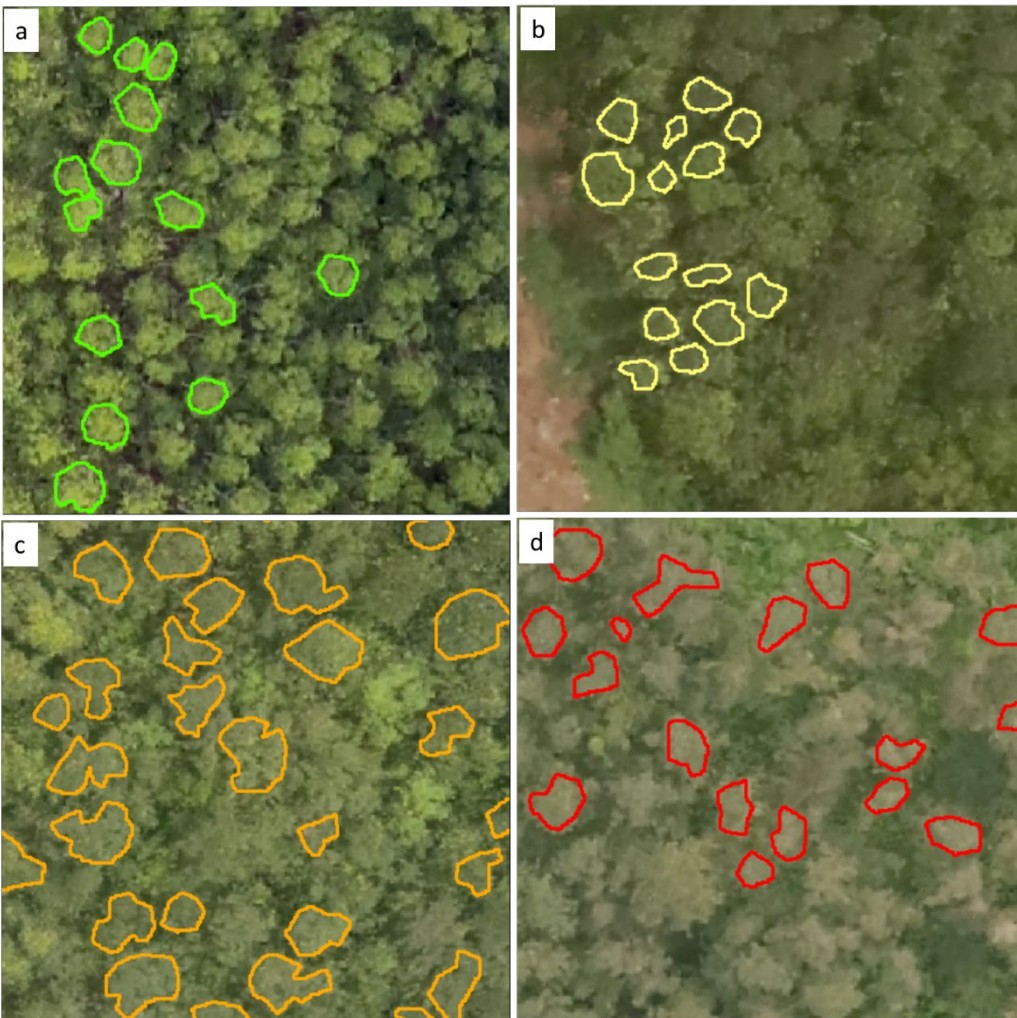

**Figure 3.** Examples of eucalyptus tree health on true-color UAV images: (**a**) healthy; (**b**) mildly infected; (**c**) moderately infected; (**d**) severely infected. The polygons are tree samples manually drawn through comparison of the field survey with UAV images.

*2.4. Identification of Optimal Variables Using a Mutual Information–Based Feature Selection Method*

Feature selection is the process of identifying important input variables while removing irrelevant and redundant ones from the originally large number in the feature set, aiming to reduce computational cost during the training process and to increase the performance of classification or regression models. Because many features can be derived from remotely sensed data (for example, spectral bands, vegetation indices, textures, and principal components), some of these features are highly correlated; not all features are equally important for a specific study. It is necessary to identify the most important features that are relevant to the research target. Mutual information–based feature selection method

(MI) has the ability to capture both linear and non-linear dependences between two (or more) random variables, is easy to compute and interpret, and has been extensively used for identifying an optimal subset of features [53–55]. MI measures the amount of information shared by two random variables and can be described as the amount of the uncertainty of one random variable reduced given the knowledge of another random variable. MI can be calculated using Equation (1):

$$MI(X,Y) = \sum_{x \in X} \sum_{y \in Y} p(x,y) \log\left(\frac{p(x,y)}{p(x)p(y)}\right) \tag{1}$$

where $p(x)$ and $p(y)$ are the marginal probability distributions of variables $X$ and $Y$, respectively, and $p(x,y)$ is the joint probability distribution of variables $X$ and $Y$. The larger the MI score, the higher the dependence between $X$ and $Y$.

This study used MI to evaluate the dependencies between the disease severity levels of eucalyptus trees and remote sensing–derived features. The features used for detecting diseased eucalyptus include UAV spectral bands and vegetation indices. This process starts with UAV spectral bands. MI scores between disease severity levels and the brightness values of spectral bands were calculated using tree samples within R package "infotheo," and the dependencies between spectral bands and health conditions were evaluated. The spectral bands with higher MI scores are considered more sensitive to health status, and were selected for further analyses. This step identified the three most sensitive bands, i.e., green, red edge, and NIR. It is assumed that the vegetation indices associated with the selected bands are also potentially sensitive to health status. Thus, common vegetation indices, which contain any one of the selected spectral bands (Table 1), were calculated. Similarly, MI scores between disease severity and vegetation indices were generated and tevaluated.

**Table 1.** Vegetation indices based on the selected spectral bands sensitive to tree health.

| Name & Abbreviation | Equation |
|---|---|
| Normalized Difference Vegetation Index, NDVI | $NDVI = (R_{nir} - R_r)/(R_{nir} + R_r)$ |
| Green Normalized Difference Vegetation Index, GNDVI | $GNDVI = (R_{nir} - R_g)/(R_{nir} + R_g)$ |
| Normalized Difference Red Edge Index, NDREI | $NDREI = (R_{nir} - R_{re})/(R_{nir} + R_{re})$ |
| ChlorophyII Index-Red Edge, CIRE | $CIRE = R_{nir}/(R_{re} - 1)$ |
| ChlorophyII Index-Green, CIG | $CIG = R_{nir}/(R_g - 1)$ |
| Anthocyanin Reflectance Index, ARI | $ARI = R_g^{-1} - R_{re}^{-1}$ |
| Nitrogen Reflectance Index, NRI | $NRI = (R_g - R_r)/(R_g + R_r)$ |
| Greenness Index, GI | $GI = R_g/R_r$ |
| Optimized Soil Adjusted Vegetation Index, OSAVI | $OSAVI = 1.16 (R_{ni} - R_r)/(R_{nir} + R_r + 0.16)$ |
| Ratio Vegetation Index, RVI | $RVI = R_{nir}/R_r$ |
| Transformed Chlorophyll Absorption and Reflectance Index, TCARI | $TCARI = 3 [(R_{re} - R_r) - 0.2 (R_r - R_g) (R_{re}/R_r)]$ |

Note: $R_{nir}$, $R_g$, $R_r$, and $R_{re}$ represent spectral reflectance of near-infrared, green, red, and red edge bands, respectively.

### 2.5. Detection of Eucalyptus Tree Health Conditions

Two classification methods (RF and SAM) were used in this study to classify eucalyptus trees into four predefined health categories based on UAV multispectral data. RF is a tree-based supervised machine learning algorithm that combines ensemble learning, bagging sampling method, and feature randomness [56]. Because it is a nonparametric algorithm, it does not need any assumptions about data distribution. The advantages of easy implementation, efficient handling of large datasets, high versatility for classification and regression tasks, high accuracy of prediction, and ability to rank feature importance make it widely used in various fields such as variable selection, land-cover classification, and forest biomass estimation [57,58].

SAM is a physically based spectral classification method that treats the multispectral data as a vector in multidimensional space (*n*-D, *n* equal to the number of spectral bands), and calculates the angle between the spectra of an unknown pixel and a labeled pixel or

an endmember to determine the degree of similarity between two spectra [59]. The angle values range between 0 and 1. The smaller the angles, the closer the spectra of unclassified pixels to the reference spectra [60,61]. The final classified image shows the best match at each pixel. The calculation of spectral angles can be expressed as Equation (2):

$$\theta = arccos \frac{T \cdot R}{|T||R|} \tag{2}$$

where $T$ is unknown spectra vector $T$ ($t_1$, $t_2$, . . . , $t_n$), $R$ is reference spectra vector $R$ ($r_1$, $r_2$, . . . , $r_n$).

This study used RF and SAM to develop tree health classification models based on 303 tree samples. The optimal spectral bands as input features were first input to RF and SAM models, followed by the combination of the optimal spectral bands and vegetation indices as input variables. Thus, four scenarios were designed: spectral bands with RF, combination of spectral bands and vegetation indices with RF, spectral bands with SAM, and combination of spectral bands and vegetation indices with SAM.

### 2.6. Accuracy Assessment and Mapping of Eucalyptus Health Conditions

Confusion matrices were created using the validation samples and corresponding classification results from the four scenarios described in Section 2.5. Producer's accuracy (PA), user's accuracy (UA), overall accuracy (OA), and kappa coefficients were calculated and used for accuracy assessment [62]. By comparing the classification accuracies of different scenarios, the optimal scenario of feature combination and classification algorithm was determined and then used to predict the health conditions of eucalyptus trees in the whole area.

## 3. Results

### 3.1. Identification of Optimal Spectral Bands and Vegetation Indices Based on MI Scores

A comparative analysis of the MI values based on different health conditions of tree samples and corresponding UAV spectral bands and vegetation indices (Figure 4) indicates that the highest MI score goes to the green band, followed by the red edge and NIR bands, and the lowest MI score is with blue, implying that green, red edge, and NIR bands are more sensitive than are blue and red bands to the disease severity levels. Thus, green, red edge, and NIR bands were chosen for discriminating between different health conditions in the classification stages; they also were used for calculating a series of vegetation indices, as listed in Table 1. Based on MI values, nitrogen reflectance index (NRI) and greenness index (GI) have the highest MI scores (>0.5) among the selected vegetation indices, implying their high sensitivity to tree health levels. Therefore, NRI and GI were selected for further analysis. Figure 4 also shows that NRI and GI have higher MI values than do any of the spectral bands, indicating they are more sensitive to tree disease levels than are spectral bands. Thus, a combination of spectral bands and vegetation index has the potential to improve the separation performance of different disease levels. Because these two vegetation indices are highly correlated (the correlation coefficient reached 0.98), use of either one should produce similar results. Thus, two data scenarios—(1) sensitive spectral bands (i.e., green, red edge, and NIR) alone, (2) a combination of sensitive spectral bands and NRI—were formed for discrimination of eucalyptus health conditions.

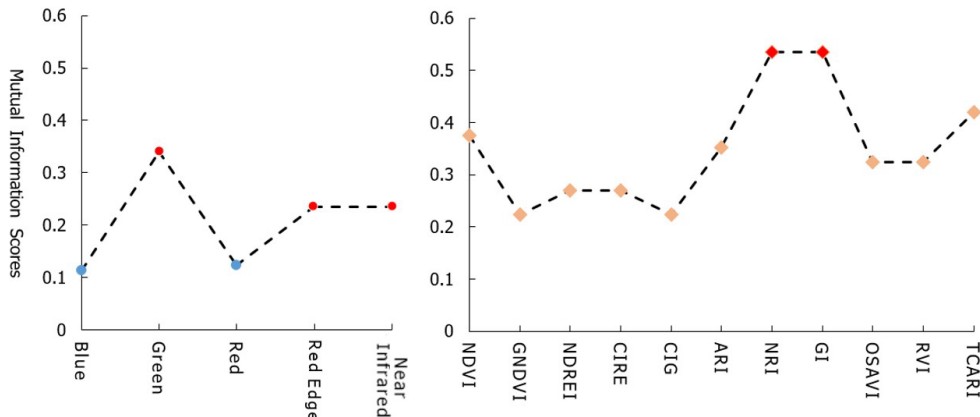

**Figure 4.** A comparison of mutual information values based on tree samples at different health conditions and corresponding UAV spectral bands (**left**) and vegetation indices (**right**). See Table 1 for full names of indices.

### 3.2. Accuracy Assessment of Eucalyptus Health Conditions Detection Results

The comparison of accuracy assessment results (Table 2) indicates that overall, the combination of the three spectral bands and the vegetation index greatly improved classification accuracies compared to spectral bands alone, using either the RF or SAM algorithm. In terms of algorithms, RF performed better than SAM given the same predictor features. As shown in Table 2, the overall accuracies and kappa values based on spectral bands alone were 84.6% and 0.79 with RF, and 71.4% and 0.62 with SAM, respectively. After incorporating NRI, the overall accuracy and kappa with the RF method increased to 90% and 0.87, respectively.

Because RF outperforms SAM in terms of overall accuracy and kappa coefficient, we focused on RF to examine the role of the vegetation index in improving detection accuracy (Table 2). When using spectral bands alone, the major errors were from misclassification of moderate level to severe level, leading to a low PA of severe level (66.7%). Incorporation of a vegetation index into spectral bands improved detection accuracy in all categories; for example, UA of the health level reached 96.8%, and UA of the severe level was greater than 90%, indicating the importance of a vegetation index in separation of healthy and infected trees.

**Table 2.** Detection accuracies of various eucalyptus tree health conditions based on different features and algorithms.

| Data and Method | Classified Levels | Reference Data | | | | | |
| --- | --- | --- | --- | --- | --- | --- | --- |
| | | **Healthy** | **Mild** | **Moderate** | **Severe** | **Total** | **UA (%)** |
| | Healthy | 28 | 1 | 0 | 2 | 31 | 90.4 |
| | Mild | 1 | 12 | 1 | 1 | 15 | 80.0 |
| | Moderate | 0 | 1 | 21 | 5 | 27 | 77.8 |
| Spectral bands with RF | Severe | 1 | 1 | 0 | 16 | 18 | 88.9 |
| | Total | 30 | 15 | 22 | 24 | 91 | |
| | PA (%) | 93.3 | 80 | 95.5 | 66.7 | | |
| | **OA (%)** | | **84.6** | | **Kappa** | | **0.79** |
| | Healthy | 30 | 1 | 0 | 0 | 31 | 96.8 |
| | Mild | 0 | 11 | 1 | 1 | 13 | 84.6 |
| | Moderate | 0 | 1 | 21 | 3 | 25 | 84.0 |
| Spectral bands & NRI with RF | Severe | 0 | 2 | 0 | 20 | 22 | 90.9 |
| | Total | 30 | 15 | 22 | 24 | 91 | |
| | PA (%) | 100 | 73.3 | 95.5 | 83.33 | | |
| | **OA (%)** | | **90.1** | | **Kappa** | | **0.87** |

**Table 2.** *Cont.*

| Data and Method | Classified Levels | Reference Data | | | | | |
| | | Healthy | Mild | Moderate | Severe | Total | UA (%) |
|---|---|---|---|---|---|---|---|
| Spectral bands with SAM | Healthy | 20 | 2 | 2 | 0 | 24 | 83.3 |
| | Mild | 5 | 10 | 1 | 1 | 17 | 58.8 |
| | Moderate | 5 | 2 | 16 | 4 | 27 | 59.3 |
| | Severe | 0 | 1 | 3 | 19 | 23 | 82.6 |
| | Total | 30 | 15 | 22 | 24 | 91 | |
| | PA (%) | 66.7 | 66.7 | 72.7 | 79.2 | | |
| | **OA (%)** | | **71.4** | | **Kappa** | | **0.62** |
| Spectral bands & NRI with SAM | Healthy | 28 | 2 | 1 | 0 | 31 | 90.3 |
| | Mild | 2 | 11 | 2 | 1 | 16 | 68.8 |
| | Moderate | 0 | 0 | 18 | 2 | 20 | 90.0 |
| | Severe | 0 | 2 | 1 | 21 | 24 | 87.5 |
| | Total | 30 | 15 | 22 | 24 | 91 | |
| | PA (%) | 93.3 | 73.3 | 81.8 | 87.5 | | |
| | **OA (%)** | | **85.7** | | **Kappa** | | **0.81** |

Note: NRI, nitrogen reflectance index; RF, random forest; SAM, spectral angle map; PA, producer's accuracy; OA, overall accuracy; UA, user's accuracy.

### 3.3. Spatial Distribution of Eucalyptus Trees with Different Health Conditions

The RF based on a combination of selected features (spectral bands and NRI) was applied to the whole study area and produced a spatial distribution of eucalyptus health levels (Figure 5). The total areas and percentages of each category are summarized in Table 3, showing that about one-third of eucalyptus trees were healthy, and the remaining two-thirds suffered disease damage at various degrees, of which moderate and severe levels account for a large portion, implying the necessity to take urgent measures to control disease expansion.

**Table 3.** Areas and percentages of eucalyptus trees at various health conditions.

| Tree Health Condition | Area (m²) | Percentage of Total Eucalyptus Area (%) |
|---|---|---|
| Healthy | 91,533 | 33.0 |
| Mildly infected | 52,274 | 18.9 |
| Moderately infected | 64,463 | 23.2 |
| Severely infected | 69,104 | 24.9 |
| Total | 277,375 | 100 |

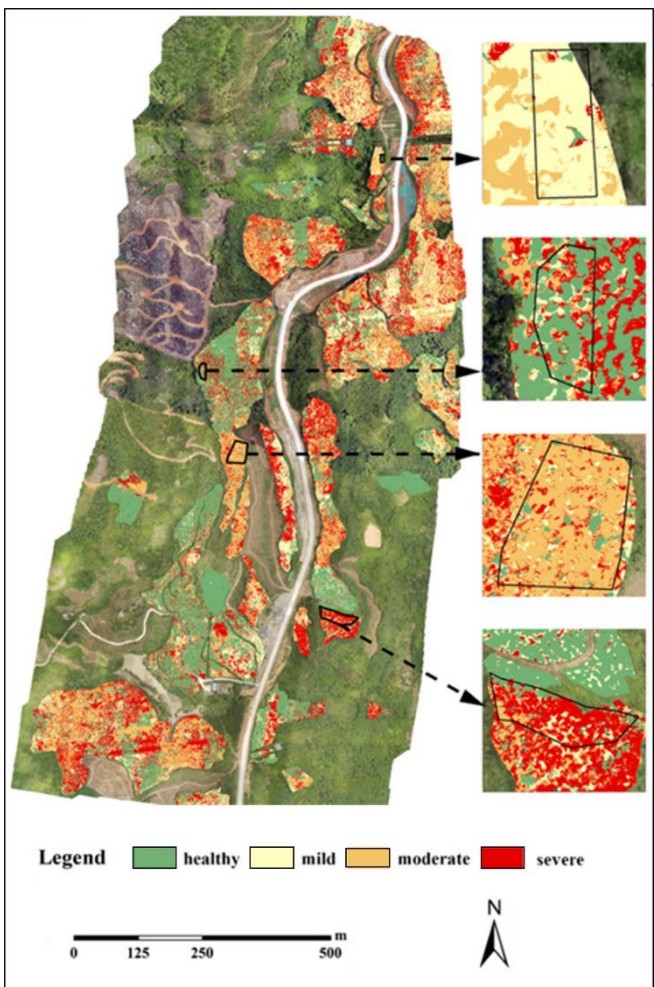

**Figure 5.** Spatial distributions of eucalyptus health conditions using random forest classifier based on the combination of three spectral bands and the nitrogen reflectance index. Tree health condition was defined according to the percentage of unhealthy leaves: 0 is healthy; 0%–30% is mild; 30%–60% is moderate; >60% is severe.

## 4. Discussion

### 4.1. The Importance of UAV Data and Feature Selection in Tree Disease Detection

This study used UAV-acquired multispectral images to detect eucalyptus leaf disease and assess its severity levels, finding that green, red edge, and NIR bands were more sensitive than were blue and red bands to infected trees and had good features for identifying different levels of infected trees, a conclusion similar to those of Kumbula et al. [18] and Oumar and Mutanga [19]. During our field survey, we found that healthy eucalyptus leaves had obviously different characteristics from infected leaves in color and water content. The healthy leaves were green and held moisture, while infected leaves were brown or dark and very dry, depending on different severity levels, as shown in Figure 3. The differences in pigment and spongy mesophyll between healthy and infected leaves made the green spectral band the most sensitive to healthy conditions, according to analysis of MI scores. This finding provides the guideline to select proper vegetation indices to improve the separation of different severity levels.

A vegetation index, which combines two or more spectral bands, is a simple and effective way to enhance vegetation characteristics, such as vigor. Because the green spectral band has higher sensitivity to healthy conditions than do other spectral bands, NRI and GI containing the green band appear to yield higher MI values than do other vegetation indices, as shown in Figure 4, implying the important role of these VIs in improving the separation

of different severity levels (see Table 2). For the eucalyptus diseases, the VARI-green index (visual atmospheric resistance index-green index) calculated from UAV-acquired RGB data was successfully used to identify the infected trees in a young eucalyptus plantation in Malaysia [5]. Another study based on spectral bands and various vegetation indices from Sentinel-2 multispectral images using the Maxent algorithm found that the normalized difference photosynthetic vigor ratio (PVR), an index that involved green and red bands, was effective for detecting eucalyptus disease [18]. The NRI and GI selected in this research are similar to the PVR, and also exhibit higher power than do other vegetation indices in detection of eucalyptus tree health status when used with spectral bands. These studies confirmed the value of using vegetation indices containing the green spectral band for detection of eucalyptus tree disease.

For high–spatial resolution images, effectively extracting the rich spatial information is often key to improving classification accuracy as to detailed tree species, and aids in incorporating this kind of information into spectral features [58,63]. In this research, we examined the potential of using spatial-based features derived by using a gray-level co-occurrence matrix approach with different window sizes for classification of different levels of infection in trees, and found that textures had little effect on improving the detection accuracy of eucalyptus leaf disease. This may be due to the fact that leaf disease mainly changes the leaf colors without causing change in the spatial relationships of the leaves. Since eucalyptus tree disease mainly induces the color change of leaves without inducing leaves to drop off, except in the extremely severe condition, spectral reflectance can effectively detect the subtle change of leaf colors, especially with green, red edge, and NIR spectral bands. This research indicates the importance of identifying suitable vegetation indices based on the spectral bands sensitive to tree disease instead of the spatial features.

### 4.2. The Necessity to Select a Suitable Algorithm to Detect Tree Disease Levels

The comparison of two algorithms showed that RF outperformed SAM when the same features were used as inputs for classification. The RF algorithm has no assumptions about data types and data distribution. However, the SAM classification requires the use of reflectance data [59]. Besides, SAM required endmember or library spectra obtained by prior experiments. The multispectral data acquired by UAV in this study have not been calibrated, and the calculated vegetation indices based on the selected spectral bands may be affected by non-reflectance data. Also, the sampled trees for various health conditions are not homogeneous. All these factors may contribute high errors in classification using SAM. Recent research has explored the application of deep learning algorithms to detect plant diseases [35,36,64–66], offering opportunities for plant disease detection [64,67,68]. More effort will be needed in the future for precise identification of eucalyptus disease using advanced deep learning methods based on high-resolution data.

### 5. Conclusions

Based on high–spatial resolution multispectral images obtained by UAV, this study identified the optimal features for discriminating healthy and infected eucalyptus trees through the MI method, and evaluated the performance of RF and SAM algorithms for distinguishing different health levels in eucalyptus trees. The results indicate that green, red edge, and NIR bands, together with the NRI index and green index, are sensitive to eucalyptus tree diseases. RF performed better than SAM, given the same feature inputs. The combination of spectral bands with a vegetation index using RF produced an overall accuracy of 90.1%. Within the study area, two-thirds of eucalyptus trees suffered disease infection, and most of them were at moderate and severe levels. The resulting map illustrates the accurate locations of infected trees, where farmers should pay more attention and take prompt measures to prevent infestation. This study demonstrated a practical way to quickly detect eucalyptus health using UAV multispectral images, exemplifying the effectiveness of UAV technology in assessing its severity in a subtropical region. The findings and results

will be valuable for local eucalyptus owners, enabling them to take appropriate measures to target the infected trees or specific areas to prevent disease expansion.

**Author Contributions:** Conceptualization, K.L. (Kuo Liao), H.D., and G.L.; methodology, F.Y., K.L. (Kuo Liao), and G.L.; software and validation, F.Y., Y.W., and K.L. (Kuo Liao); formal analysis, F.Y. and G.L.; investigation, F.Y., Y.W., and K.L. (Kunfa Luo); resources and data curation, K.L. (Kuo Liao), H.D., and K.L. (Kunfa Luo); writing—original draft preparation, F.Y. and G.L.; writing—review and editing, K.L. (Kuo Liao), and G.L.; visualization, F.Y. and G.L.; supervision, project administration, and funding acquisition, K.L. (Kuo Liao), and G.L. All authors have read and agreed to the published version of the manuscript.

**Funding:** This research was funded by Fujian Provincial Department of Sciences and Technology, grant number 2021R1002008.

**Data Availability Statement:** Data are available on request.

**Acknowledgments:** The authors would like to thank Dengqiu Li and Yaoliang Chen for their contributions in field data collection and discussion of this research work.

**Conflicts of Interest:** The authors declare no conflict of interest. The funders had no role in: design of the study; collection, analyses, or interpretation of data; writing of the manuscript; or the decision to publish the results.

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
