# Peer review of "Detection of Eucalyptus Leaf Disease with UAV Multispectral Imagery"

_forests, doi:10.3390/f13081322_

Round 1

Reviewer 1 Report

This paper provides a useful guide to disease detection using UAVs over Eucalyptus trees and recommends that additional tree species be tested and that indices work better then single bands. 

The methods used are well documented and known methods and are applied appropriately. 

Results are suitable presented and discussed.

The main issue I have is with the grammar. Many sentences need to be restructured. I stopped making corrections after the first paragraph as the problems are quite extensive. Once this is corrected I believe you will have a good paper. 

Please see attached for further comments. 

Author Response

Thank you very much for grammar corrections and other great comments and suggestions. According to your suggestions, we made modification.

  1. Grammar errors:

Reply: According to your suggestion, we corrected them all. Also, the manuscript was checked by a professional English editor Joanna T. Broderick

  1. Key words:

Reply: We changed key words to “Eucalyptus diseases; mutual information-based feature selection; random forest; spectral angle mapper; vegetation index” according to your suggestion.

  1. Line 112. If there is a treatment for these diseases it would be good to mention this. UAV's promote early diagnosis and treatment and therefore reduced timber losses? If there is no treatment then UAVs just tell us how much we have lost?

Reply: This is the great suggestion. The early diagnosis will help find infected areas, where the appropriate treatments and managements should be focused on to control further damage, reducing the loss caused by disease. Thus, we added typical methods for eucalyptus disease and insect treatment and management in the text. “Chinese researchers have developed various treatments and managements to prevent and control eucalyptus diseases, including pruning or removing infected parts or whole trees, spraying with pesticides, introducing biological enemies, improving soil conditions, and planting disease-resistant clones. Accurate detection of infected trees and assessment of infection severity are prerequisite for implementing those measures promptly. Remote sensing has long been used to detect eucalyptus crown damage caused by insects and diseases”

  1. Page4 line 159 delete “Super high spatial resolution”

Reply: “Super high spatial resolution” was deleted.

  1. Figure 2 Does this need to connect to the Random forest as well?

Reply: We modified the flowchart and corrected it.

  1. Figure 3 Is the reader meant to be able to see the difference in tree health? I"m not sure this is obvious to the reader? It is not clear whether the tree health described here is according to on site classification or UAV classification. What are the polygons around the canopies? They are not clear and not described.

Reply: Thank you for the comments. We are sorry for not being able to provide the perfect pictures to show the typical appearances of eucalyptus tree health conditions. Ideally, we should take pictures from above canopy or close looking to show the leaf colors and the amount of unhealthy leaves. However, the eucalyptus trees are so high for taking clear pictures. We tried our best to make clearer images, but not successful. Thus, we enlarged the images, making it a little better. The polygons on the top of image are the tree samples manually drawn on UAV images based on the field survey.

  1. Line 180 Please give examples of what kind of features these are.

Reply: In text, we added examples of remote sensing derived features, such as spectral bands, vegetation indices, texture, principal components, etc.

  1. Line 222 specify that this is the UAV imagery you are classifying

Reply: We specified “based on UAV multispectral data” 

  1. Figure 5 Probably need to specify that mild, mod and severe refer to extent of disease

Reply: We added the definitions of health conditions in caption

  1. Table 3 “Healthy conditions” to Tree condition

Reply: We revised Table 3, making it clearer.

  1. Line 316 I suggest that you rather say the diseased trees could be identified more accurately using vegetation indices. Rather don't say that the spectral banks are not efficient for accuracy.

Reply: This is a great suggestion. We modified the sentences according to your comments.

  1. Line 319 Explain why you included this topographic and atmos influences here.

Reply: Thank you very much for the suggestion. This section is considerably revised by focusing on spectral bands and vegetation indices, and removing the contents related to topographic and atmospheric issues. Based on our field survey, the infected trees are different from the healthy ones in color and moisture, which are related to spectral values; vegetation index is an important indicator of vegetation health, and was proven effective in diseased trees identification.”

  1. Line 320 I"m not sure this fully captures the difference? Different amounts of chlorophyll etc?

Reply: this part is re-written.

  1. Line 351 This last sentence in not that useful. You didn't use hyperspectral so how can you say that more research is needed. Based on the findings of others, hyperspectral would give greater accuracies but the issue is the price?

Reply: This is good point. We deleted this paragraph related to hyperspectral data.

  1. Line 366 Can you suggest why this may be the case?

Reply: We deleted this part because we did not explore the use of hyperspectral data in this research. 

Reviewer 2 Report

1. You should clearly mention the exact area of you study site in km2 or hectares in the study area.

In line 158 you mentioned:"experimental site covering about 1 km2" and the sum of the areas in Table 3 is 277375 m2 (0.277 km2).

Clarification should be made.

2. It's very informative if you show the boundary of areas where the eucalyptus tresses are in Figure 1. Because the reader imagines that the eucalyptus trees are in all parts of area. But in the fig 5 it showed that eucalyptus tresses are existed in some parts.

Author Response

We thank you for the great suggestions. According to your suggestions, we made clarification and created a new map.

  1. You should clearly mention the exact area of you study site in km2 or hectares in the study area.  In line 158 you mentioned: "experimental site covering about 1 km2" and the sum of the areas in Table 3 is 277375 m2 (0.277 km2). Clarification should be made.

Reply: Line 158 “experimental site covering about 1 km2” means the area UAV flew over or multispectral image covers; Table 3 lists the areas of Eucalyptus plantations at different health conditions, the total eucalyptus area is 0.277 km2. Also, this number was added into Table 3.

  1. It's very informative if you show the boundary of areas where the eucalyptus tresses are in Figure 1. Because the reader imagines that the eucalyptus trees are in all parts of area. But in the fig 5 it showed that eucalyptus tresses are existed in some parts.

Reply: We created a new map (Figure 1), which shows the eucalyptus plantation distribution within the experimental site.

Reviewer 3 Report

This study used UAV-acquired multispectral images to detect eucalyptus leaf disease and assess the severity levels, indicating that green, RedEdge, and NIR bands were more sensitive to infected trees and had good features for identifying different levels of infected trees. Different variables – spectral bands and vegetation indices were used to distinguish forest disease levels using random forest and spectral angle mapper approaches, respectively. The results show that green, red, and near-infrared wavelengths, nitrogen reflectance index, and greenness index are sensitive to forest diseases.

Concept Comments

In this research, authors examined the potential of using spatial-based features using gray-level co-occurrence matrix approach with different window sizes for classification of different levels of infected trees. Authors found that textures had little effects on improving the detection accuracy of eucalyptus leaf disease. This may be due to the fact that the leaf disease only changes the leaf colors without causing the spatial relation-ships of leaves. Since eucalyptus tree disease mainly induces the color change of leaves without inducing leaves off except the extremely severe condition, spectral reflectance can effectively detect the subtle change of leaf colors, especially green, RedEdge, and NIR spectral bands. This research indicates that identification of suitable vegetation indices based on the sensitive spectral bands to tree disease is more important than the spatial features.

Weakness and suggestions for improvement:

1.   Hyperspectral images were not used for different temporal characteristics which could have improved the accuracy.

2.   Authors used various combinations of Spectral bands and NRIs but did not use NRIs alone. The experimental values obtained from NRIs may also be depicted in Table 2. If not possible, the reason for the same may also be mentioned.

Author Response

  1. Hyperspectral images were not used for different temporal characteristics which could have improved the accuracy.

Response: yes, hyperspectral images were not used in this research, thus, the discussion contents related to hyperspectral images were removed in this revised paper.

  1. Authors used various combinations of Spectral bands and NRIs but did not use NRIs alone. The experimental values obtained from NRIs may also be depicted in Table 2. If not possible, the reason for the same may also be mentioned.

Response: Thanks for your comment. As a comparison, NRI alone should be used. However, considering the fact that a single vegetation index is not as good as multispectral bands because the vegetation index is derived from a couple of spectral bands, thus, NRI alone was not used, instead, it combined into spectral bands to check whether the results can be improved because of the use of NRI.

Reviewer 4 Report

·         Abstract: line 20: “green ,red and near-infrared wavelengths are sensitive…..”. however results and conclusions shows its green, rededge and near-infrared wavelengths are sensitive. The use of feature selection method is not mentioned in abstract.

·         Feature extraction is mentioned in line 152. However it’s not explained anywhere in text.

·         Section 2 describes the area/ field where the image acquisition was performed. This section must be moved in section 3 as subsection.

·         Also, it will be appreciated if the authors mention tree density in their selected area of study.

·         What are field survey sample mentioned in section 3? Are they a part of multispectral imagery? The same should be explained in Figure 2.

·         The overall accuracy with RF is 90.1 % in abstract and 90% in conclusion? Be consistent.

·         Figure 4: MI scores are plotted versus bands and spectral indices. Its better if it’s done in separate figures with x-label marked properly.

·         How the spatial distribution of eucalyptus health conditions shown in figure 5 is obtained?

·         What is the purpose of block “design of scenario” in Figure 2? It should be a classifier either RF or SAM. Also training samples have only been fed to SAM classifier not RF? Is it so? From the figure it seems that the classification results from both classifiers are jointly evaluated in terms of accuracy metrics (as the output arrows are being combined as one). Please correct the confusion.

·         Line 317 needs correction.

·         The first paragraph of section 5 explains the importance of combining vegetation index with spatial band for disease identification. I guess it should be a part of related work as it gives a justification for your proposed method.

·         The second paragraph of section 5 explains the advantages of using hyperspectral imaging for accurate disease identification. Which is understood. But the authors must explain the limitations that made them use multispectral imagery instead of hyperspectral and are there any disadvantages associated with the later.

·         Table 3 , last column represents the % of what?

·         Figure 3 is not cited in text.

Author Response

We thank you for the great comments and suggestions. According to your suggestions, we made intensive revision.

  1. Abstract: line 20: “green ,red and near-infrared wavelengths are sensitive…..”. however results and conclusions shows its green, rededge and near-infrared wavelengths are sensitive. The use of feature selection method is not mentioned in abstract.

Reply: We are sorry for such careless mistake. The right statement should be “green, RedEdge, and near-infrared wavelengths are sensitive…”. We corrected it. Also, in Abstract, we included “feature selection method” in the sentence “The key variables sensitive to eucalyptus leaf diseases including spectral bands and vegetation indices were identified by using mutual information-based feature selection method, then used to distinguish disease levels using random forest and spectral angle map-per approaches, respectively”

  1. Feature extraction is mentioned in line 152. However it’s not explained anywhere in text.

Reply: We modified the whole paragraph in section 2.2 Study Framework, and deleted “feature extraction”.

  1. Section 2 describes the area/ field where the image acquisition was performed. This section must be moved in section 3 as subsection.

Reply: We moved section “Study Area” into “Section 2 Materials and Methods” as a subsection.

  1. Also, it will be appreciated if the authors mention tree density in their selected area of study.

Reply: Yes. We added more information about eucalyptus plantations within the experimental area, including area, tree density (1650 trees per ha), and other major tree species.

  1. What are field survey sample mentioned in section 3? Are they a part of multispectral imagery? The same should be explained in Figure 2.

Reply: We are sorry for the confusion. We made clarification about “field survey sample”. It refers to the collection of eucalyptus tree samples for different health conditions in the field.  We added more description in data collection subsection. We also modified Figure 2.

  1. The overall accuracy with RF is 90.1 % in abstract and 90% in conclusion? Be consistent.

Reply: We corrected 90% to 90.1%.

  1. Figure 4: MI scores are plotted versus bands and spectral indices. Its better if it’s done in separate figures with x-label marked properly.

Reply: We made two separate figures, one for spectral bands, one for vegetation indices.

  1. How the spatial distribution of eucalyptus health conditions shown in figure 5 is obtained?

Reply: The mapping unit is the same as multispectral image, i.e. 0.10 m.

  1. What is the purpose of block “design of scenario” in Figure 2? It should be a classifier either RF or SAM. Also training samples have only been fed to SAM classifier not RF? Is it so? From the figure it seems that the classification results from both classifiers are jointly evaluated in terms of accuracy metrics (as the output arrows are being combined as one). Please correct the confusion.

Reply: Sorry for the confusion. We revised Figure 2. The block “design of scenario” was deleted. “Training samples” links to both RF and SAM. The classification results from combination of classifier and selected variables were evaluated separately and compared. 

  1. Line 317 needs correction.

Reply: This paragraph is re-written.

  1. The first paragraph of section 5 explains the importance of combining vegetation index with spectral bands for disease identification. I guess it should be a part of related work as it gives a justification for your proposed method.

Reply: Thank you for the comments. We realized this part is not only about the importance of vegetation index, also the selection of UAV data and features for disease detection. Thus, we modified the title of this subsection. The whole section was revised intensively.

  1. The second paragraph of section 5 explains the advantages of using hyperspectral imaging for accurate disease identification. Which is understood. But the authors must explain the limitations that made them use multispectral imagery instead of hyperspectral and are there any disadvantages associated with the later.

Reply: This part related to hyperspectral data was removed because we did not use hyperspectral data in this research.

  1. Table 3, last column represents the % of what?

Reply: It is the percentage of each health category accounted for the total area of eucalyptus plantations. We modified Table 3.

  1. Figure 3 is not cited in text.

Reply: Figure 3 was cited in subsection “data collection”
